# Bootstrap Your Object Detector via Mixed Training

**Mengde Xu**[1,3*]    **Zheng Zhang**[1,3*]    **Fangyun Wei**[3*]    **Yutong Lin**[2,3*]    **Yue Cao**[3]

**Stephen Lin**[3]    **Han Hu**[3]    **Xiang Bai**[1]

[1]Huazhong University of Science and Technology
[2]Xi'an Jiaotong University
[3]Microsoft Research Asia

## Abstract

We introduce MixTraining, a new training paradigm for object detection that can improve the performance of existing detectors for free. MixTraining enhances data augmentation by utilizing augmentations of different strengths while excluding the strong augmentations of certain training samples that may be detrimental to training. In addition, it addresses localization noise and missing labels in human annotations by incorporating pseudo boxes that can compensate for these errors. Both of these MixTraining capabilities are made possible through bootstrapping on the detector, which can be used to predict the difficulty of training on a strong augmentation, as well as to generate reliable pseudo boxes thanks to the robustness of neural networks to labeling error. MixTraining is found to bring consistent improvements across various detectors on the COCO dataset. In particular, the performance of Faster R-CNN [24] with a ResNet-50 [13] backbone is improved from 41.7 mAP to 44.0 mAP, and the accuracy of Cascade-RCNN [1] with a Swin-Small [22] backbone is raised from 50.9 mAP to 52.8 mAP.

## 1    Introduction

Object detection is a fundamental task of computer vision. Its goal is to locate the bounding boxes of objects in an image as well as to classify them. Due to the complexity and diversity of the real world, this problem remains challenging despite the considerable attention it attracts. Most previous works focus on developing better detection frameworks [24, 34, 5, 2, 21, 23, 16] or stronger network architectures [18, 1, 14, 28, 12, 14]. In addition, there are some that study data augmentation strategies [10, 6, 4], label assignment [39, 17], or training losses [19, 25]. In general, the existing works follow a standard training paradigm where the network takes training images that are augmented by a *single* data augmentation strategy and the human-annotated bounding boxes are *simply* used as the training targets. We refer to this approach as *SiTraining*. Few works explore alternative training methods, which have been based on distillation [3, 32, 40] or dynamic adjustment of label assignment criteria and regression loss [35].

In this work, we expand the power of an augmentation strategy and the utility of human-annotated boxes through a new training paradigm for object detection. We observe that suitable magnitudes of an augmentation can vary from image to image, where a strong augmentation of certain images will enrich the training data, but may degrade the data when applied to other images, such as by altering the object appearance to become less compatible with the object class. Based on this, we present a mixed augmentation strategy that utilizes both strong and normal augmentations in a manner that takes advantage of strong augmentations only on images where they are expected to be helpful.

---

*Equal contribution.

35th Conference on Neural Information Processing Systems (NeurIPS 2021).

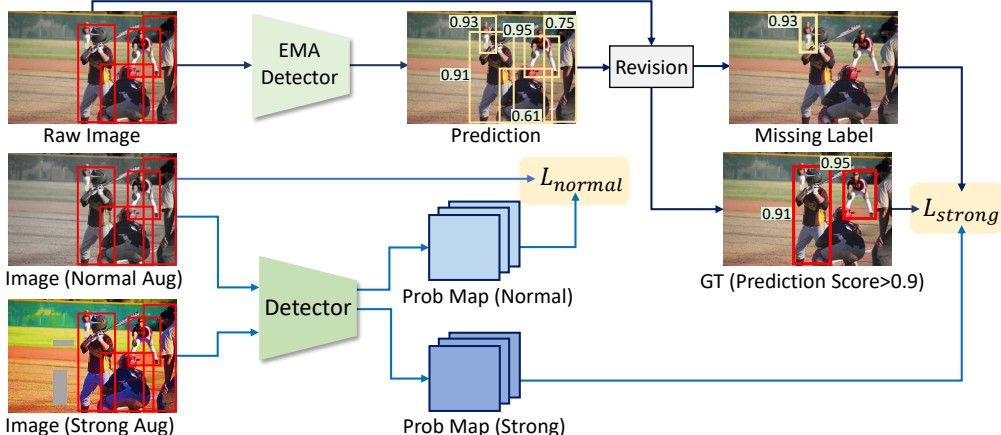

Figure 1: The illustration of our training paradigm *MixTraining*. It integrates mixed augmentation and mixed training targets. In the bottom two branches, normally augmented images and strongly augmented images are passed to the detectors for training. In the top branch, an EMA detector is used to generate pseudo boxes and predict the foreground scores of the training targets. Only targets with a score higher than 0.9 will be used for training strongly augmented images.

We additionally note that human annotation of bounding boxes is often noisy or incomplete, which can be harmful to training. To alleviate this issue, we introduce the use of mixed training targets, composed of both human-annotated ground-truths and pseudo boxes that are intended to compensate for annotation errors.

The pseudo boxes are determined by bootstrapping on the detector. Because of the robustness of neural networks to label noise, the online detector can produce pseudo boxes that capture object locations missed or inaccurately localized by human labelers. We therefore utilize these pseudo boxes in conjunction with the human-annotated boxes to improve detection. Furthermore, the online detector can predict whether a strong augmentation of a training image would help training. This is accomplished by using the online detector to compute the foreground score of a training target. A high score indicates that the target can be easily trained on, while those with a low score are discarded from training.

Our training paradigm, called *Mix-Training*, integrates the mixed augmentation and mixed training targets as illustrated in Figure 1. In the bottom two branches, normally augmented and strongly augmented images are passed to the detector for training. In the top branch, an exponential moving average (EMA) model of the online detector is used to generate pseudo boxes and to predict the foreground scores of the training targets. Only for training targets with a high score will their strong augmenta-

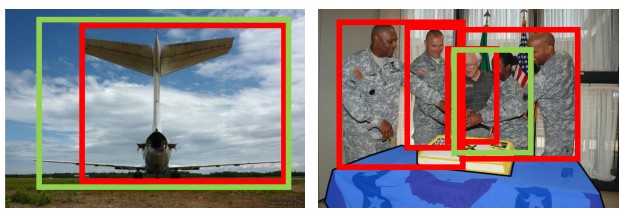

Figure 2: Illustration of annotation noise in COCO2017. (left) The red box is an inaccurate annotation, and the correct localization is the green box. (right) Red boxes are original annotations, and the green box is the missing label.

tions be included for training. As the detector progresses through training, the mixed augmentation and mixed training targets become increasingly better via bootstrapping.

*MixTraining* is a general training framework for object detection that can enhance existing object detectors without introducing extra computation or model parameters at inference time. Our experiments show that *MixTraining* can appreciably improve the performance of leading object detectors such as Faster R-CNN [24] with a ResNet-50 [13] backbone (from 41.7 mAP to 44.0 mAP) and Cascade R-CNN [1] with the recently proposed Swin-Transformer [22] backbone (from 50.9 mAP to 52.8 mAP).

## 2 Related works

**Framework and Network Design in Object Detection**   Designing more effective frameworks has been a research focus in object detection. Most object detectors can be separated into two framework categories: single-stage object detectors [21, 23, 19, 29] and two-stage object detectors [24, 5, 16]. Although we mainly conduct experiments of our method on two-stage detectors to verify the effectiveness this work, as a general training paradigm, *MixTraining* is suitable for both single-stage and two-stage detectors.

There are also many works exploring stronger network architectures. For example, feature pyramid network [12] utilizes a pyramid network to tackle the challenge of scale diversity, and Cascade R-CNN [1] introduces a multi-stage head for improving localization accuracy. These methods can significantly elevate detection performance, and our work is compatible with these methods.

**Data Augmentation in Object Detection**   Recently, the importance of data augmentation has become evident in the research community. Existing works can be classified into two categories. Methods in the first category focus on developing new transformation components. For example, Mixup [36] proposes to use mixed images and labels as training samples. CutOut [8] masks a part of a region in the original inputs. CopyPaste [11] and InstaBoost [10] further consider data augmentation at the instance level, but both of them require instance mask annotations. Another type of method studies how to combine existing transformations more effectively. Representative works include AutoAug [41] and RandAug [6]. Different from those methods that focus on how to improve a single augmentation strategy, our method considers two augmentations of different magnitudes. Current augmentation methods are, in fact, compatible with our mixed augmentation strategy, which can be applied with arbitrary forms of augmentation.

**Label Assignment in Object Detection**   Existing works on label assignment mainly focus on improving assignment accuracy. GuidedAnchoring [30] dynamically changes the shape of the anchor and leverages semantic features to fit the objects better. ATSS [37] presents an adaptive label assignment mechanism by dynamically adjusting the IoU threshold. AutoAssign [39] assigns positive or negative weights for each sample based on the training loss. Different from those methods, *MixTraining* does not change the mechanism of label assignment itself, but changes the availability of each training target based on its predicted foreground score and the magnitude of the data augmentation.

**Alternative Training Methods for Object Detection**   Only a few works explore alternative training methods for object detection, and they have primarily been based on distillation. For example, in [32, 40], feature distillation is presented to obtain fine-grained features or reduce invalid contextual information. Another work [38] focuses on the localization ambiguity issue, which is also explored in this work but we arrive at a different conclusion. In [3], the distillation of features and detection results are considered at the same time. However, they rely on a well-trained teacher detector and mainly focus on distilling knowledge from a stronger teacher model to a weaker student detector. Compared with these distillation-based methods, *MixTraining* also consists of two branches but mainly focuses on annotation noise and measuring the training difficulty of training targets.

Dynamic R-CNN [35] examines the existing training paradigm from the training dynamics perspective. During training, it dynamically adjusts the label assignment criteria and the parameters of the regression loss for fuller training of the network. In comparison, *MixTraining* enhances training by gradually introducing new well-trained samples into the set of strongly augmented images.

## 3 MixTraining

In this section, we introduce our new training scheme, *MixTraining*, which integrates the two major components: mixed training targets and mixed data augmentation. The whole training paradigm is illustrated in Figure 1. In the following, we will introduce each component in detail.

**Mixed Training Targets**   Conventional *SiTraining* uses only human-annotated ground-truths to train the detector. However, human annotation is imperfect and often includes annotation noise such as inaccurate localization or missing labels which can harm training. Examples are shown in Figure 2.

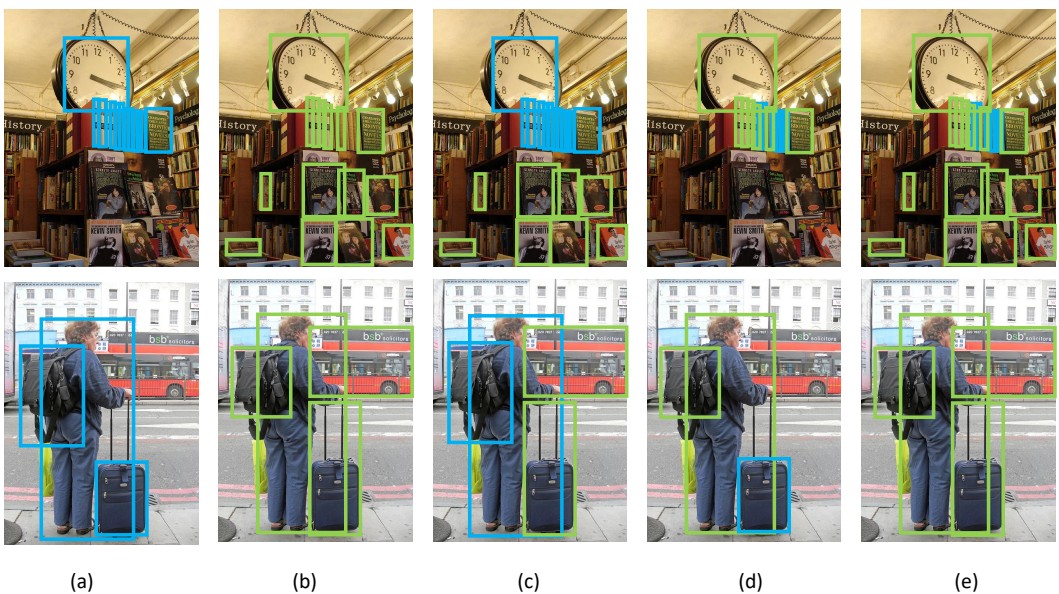

|     |     |     |     |     |
| :-: | :-: | :-: | :-: | :-: |
| (a) | (b) | (c) | (d) | (e) |

Figure 3: (a) Human-annotated ground-truth boxes (blue boxes); (b) Predicted pseudo boxes (green boxes); (c) Mixed training targets that address "Missing Label" issue; (d) Mixed training targets that address "Box Localization Noise" issue; (e) Mixed training targets that applying hybrid approach.

Because of the robustness of deep neural networks to label noise [26], they can predict pseudo boxes, also known as pseudo labels, that are relatively free of localization noise and resilient to missing labels. This property has been exploited in semi-supervised object detection [27], weakly-supervised object detection [9], and sparsely annotated object detection [31]. However, its value in supervised object detection has rarely been explored. In this work, we introduce the use of mixed training targets that include both human-annotated ground-truth boxes and pseudo boxes to alleviate the issue of noisy annotation.

To generate high-quality pseudo boxes, we follow the common practices [27, 33] of applying an exponential moving average (EMA) model on the input images without any data augmentation except for scale jitter. Since the predicted boxes of an object detector are often noisy and redundant, non-maximum suppression (NMS) is employed to eliminate redundancy, where only pseudo boxes with a foreground score greater than 0.9 are retained.

We next examine how to properly combine the pseudo boxes predicted by the object detector with human-annotated ground truth. For this, we design three strategies to deal with the missing label and noisy box localization issues:

- *Missing labels.* In this strategy, we retain pseudo boxes whose IoU (intersection-over-union) with the ground truth is less than 0.5. They can be considered as missing annotations and are added to the training targets. An example is shown in Figure 3 (c).

- *Box localization noise.* Due to occlusion, small object size, and human subjectivity, the annotation quality for box localization is often inconsistent. We alleviate this issue by replacing the ground-truth boxes by pseudo boxes whose IoU with ground-truth boxes is greater than 0.5. An example is shown in Figure 3 (d).

- *Hybrid Approach.* The above two strategies are compatible with each other, and can be used at the same time, we named this strategy as hybrid approach, which has the advantages of the above two strategies. An example is shown in Figure 3 (e).

We test the three strategies on the COCO dataset [20] in Sec. 4.3. The results show that this treatment of missing labels notably improves detection performance, and there is no significant affects that applying the strategy for box localization noise alone. However, the hybrid strategy, which takes the advantages of other two strategies, show better performance than each of the above two strategies. By thus, we adopt the hybrid strategy in producing mixed training targets.

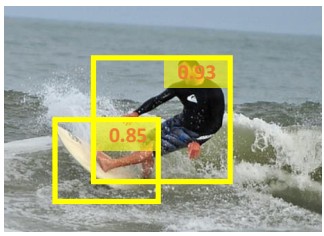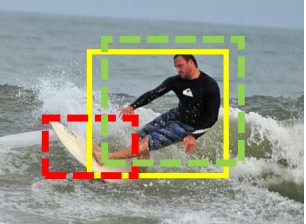

Figure 4: The boxes with solid line are training targets, and the boxes with dashed line are proposals. (Left) There are two training targets: a person and a surfboard, and their foreground score are 0.9 and 0.85, respectively. (Right) If we simply remove the surfboard (non-easy target) from the training target set, then the red proposal will be assigned as a background, because its IoU with the person is less than 0.5.

**Mixed Data Augmentation** Appropriate data augmentation is essential in training an object detector. In *SiTraining*, a single data augmentation strategy is typically utilized, and the augmentation hyper-parameters need to be carefully tuned to achieve good detection performance. However, setting suitable hyper-parameters is a challenge. Weak data augmentation is prone to over-fitting and the model capacity may not be fully utilized. On the other hand, strong data augmentation may be detrimental, as it can lead to difficulty in training for some images and training targets.

In *MixTraining*, we expand single data augmentation to a mixed strategy: during training, we randomly sample and apply an augmentation from among strong and normal augmentations for each image. Training over various levels of augmentation can be advantageous because of the greater diversity of data. However, strong augmentations can potentially hurt training if the resulting appearance becomes incongruous with object class. We address this issue by using a training loss in which strongly augmented images are taken only for targets that can be easily trained on.

A direct approach to implement this idea is to simply remove non-easy targets from all the training samples. However, a consequence of this is that many foreground proposals would be incorrectly assigned as background in the label assignment, as shown in Figure 4. Therefore, we instead assign a weight $w$ for each training target $g$:

$$w(g) = \begin{cases} 1, & g \text{ is an easy target or a normal augmentation} \\ 0, & \text{otherwise.} \end{cases} \tag{1}$$

For all images that are normally augmented, $w(g)$ is set to 1 for all of their training targets. For the image that are strongly augmented, $w(g)$ is set to 1 only if $g$ is an easy training target. Then, we use this weight $w$ as a loss weight for each proposal in the training stage:

$$L_{det} = \sum_{i=0}^{N} w(g_i) L_{det}(p_i, g_i) \tag{2}$$

where $p_i$ is the i-th proposal, and $g_i$ is the training target assigned to $p_i$.

To determine whether $g$ is an easy target, we employ a simple approach where the foreground score of $g$ is predicted by the EMA model, and $g$ is considered an easy target if its score is greater than a threshold (0.9 by default). Since this method uses the same model and input images as pseudo box generation, the extra computational overhead can be largely reduced by treating the targets $\{g_i\}$ as proposals in the pseudo box generation process.

## 4 Experiments

### 4.1 Experimental Settings and Implementation Details

**Dataset and Evaluation Protocol** We validate our method on the COCO2017 dataset [20], which contains 80 object categories, 118k images for training (train2017), 5k images (minival) for validation and 20k images for testing (test-dev). The mean average precision (mAP) is adopted as the default metric for measuring performance. In our experiments, we mainly conduct experiments on the validation set for both system-level comparison and ablation study.

**Implementation Details** In our experiments, two different data augmentation strategies are adopted: normal augmentation and strong augmentation. We summarize the details in Table 1. Compared with normal augmentation, the strong augmentation includes greater spatial/geometric transformation.

|  | Normal Augmentation | Strong Augmentation |
|---|---|---|
| Scale jitter | short edge $\in (0.5, 1.5)$ | short edge $\in (0.5, 1.5)$ |
| Solarize jitter | p=0.25, ratio $\in (0, 1)$ | p=0.25, ratio $\in (0, 1)$ |
| Brightness jitter | p=0.25, ratio $\in (0, 1)$ | p=0.25, ratio $\in (0, 1)$ |
| Contrast jitter | p=0.25, ratio $\in (0, 1)$ | p=0.25, ratio $\in (0, 1)$ |
| Sharpness jitter | p=0.25, ratio $\in (0, 1)$ | p=0.25, ratio $\in (0, 1)$ |
| Translation | - | p=0.3, translation ratio $\in (0, 0.1)$ |
| Rotate | - | p=0.3, angle $\in (0, 30°)$ |
| Shift | - | p=0.3, angle $\in (0, 30°)$ |
| Cutout | - | num $\in (1, 5)$, ratio $\in (0.05, 0.2)$ |

Table 1: Summary of the transformations we used in normal augmentation and strong augmentation. "-" indicates that the augmentation is not used.

For updating the EMA model that predicts pseudo boxes, the momentum coefficient is set to 0.999. To examine the effectiveness of *MixTraining*, we conduct experiments on various object detectors and backbone architectures. In practice, we find that *MixTraining* can benefit from a longer training schedule, while the performance of other models may degrade due to over-fitting (we discussed in Sec. 4.2). For a fair comparison, we use multiple training schedules for each model and report its best performance in our experiments. For models using ResNet-50 backbone, we adopt the SGD (stochastic gradient descent) as the default optimizer, and for models using Swin-Small as the backbone, we adopt the AdamW [15] as the optimizer. Besides, all the backbone weights are pre-trained on ImageNet-1K dataset [7]. All the models run on 32×Nvidia V100. For other training settings and hyper-parameters for each detector, we following the default settings if not otherwise specified.

| models | backbone | method | mAP | mAP@0.5 | mAP@0.75 |
|---|---|---|---|---|---|
| Faster R-CNN | ResNet-50 | *SiTraining* | 41.7 | 62.8 | 45.6 |
| | | *MixTraining* | 44.0 | 64.9 | 47.9 |
| | | $\Delta$ | +2.3 | +2.1 | +2.3 |
| Faster R-CNN | Swin-S | *SiTraining* | 48.7 | 70.5 | 53.6 |
| | | *MixTraining* | 50.3 | 71.6 | 55.2 |
| | | $\Delta$ | +1.6 | +1.1 | +1.6 |
| Cascade R-CNN | Swin-S | *SiTraining* | 50.9 | 70.3 | 55.6 |
| | | *MixTraining* | 52.8 | 72.1 | 57.9 |
| | | $\Delta$ | +1.9 | +1.8 | +2.3 |

Table 2: *SiTraining* vs. *MixTraining* on the COCO2017 validation set. Across various object detectors and backbones, *MixTraining* consistently outperforms *SiTraining* by a large margin.

| Method | 180K | 360K | 540K | 720K |
|---|---|---|---|---|
| *SiTraining* (Normal) | 41.7 | 41.7 | 40.2 | 36.1 |
| *SiTraining* (Strong) | 40.1 | 40.7 | 40.5 | 38.9 |
| *MixTraining* | - | 42.4 | - | 44.0 |

Table 3: Performance of *SiTraining* and *MixTraining* on different training iterations.

## 4.2 Comparison to *SiTraining*

***MixTraining* Benefits from Longer Training** We found that *MixTraining* can benefit from a longer training schedule due to the use of the mixed data augmentation. As shown in Figure 5 and Table 3, by extending the training iterations from 360k to 720k, *MixTraining* can improve Faster R-CNN with ResNet-50 backbone from 42.4 mAP to 44.0 mAP. On the contrary, extending the training schedule of *SiTraining* with normal data augmentation results in performance degradation because of over-fitting. Although using strong augmentation in *SiTraining* can alleviate the over-fitting issue, its best performance is worse than that for normal data augmentation because of difficulty in training.

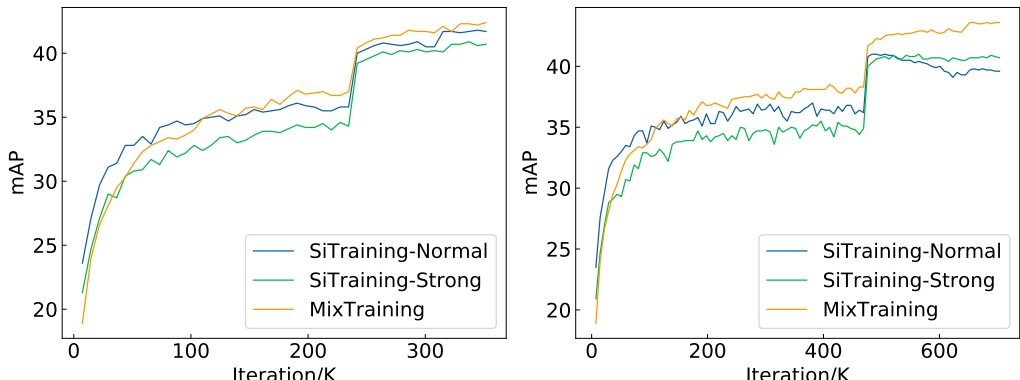

Figure 5: We illustrate the validation accuracy (mAP) of different models during training. All models shown in the left figure are trained over 360K total iterations (short training schedule), and models shown in the right figure are trained over 720K total iterations (long training schedule). The model trained by *SiTraining* (normal augmentation) is heavily over-fitted in the long training schedule. Although the use of strong augmentation can alleviate the over-fitting issue, its performance is worse than the model trained by *SiTraining* (normal augmentation) in the short training schedule. In comparison, our *MixTraining* achieves better performance while avoiding over-fitting.

To fairly compare different models, in all the following experiments, we report the best performance of different models by training the models under various training schedules.

**Comparison on Different Detectors**    We also compare *MixTraining* to *SiTraining* using different detectors. The results are shown in Table 2. When using Faster R-CNN to evaluate the two methods, *MixTraining* outperforms *SiTraining* by 2.3 points with the ResNet-50 backbone and by 1.6 points with the Swin-S backbone, which demonstrates that our method is compatible with different backbone architectures. We further conduct experiments on Cascade R-CNN to validate our method on a stronger detector, and consistent improvements are achieved, with *MixTraining* outperforming *SiTraining* by 1.9 points with the Swin-S backbone.

| Mixed Data Augmentation | Mixed Training Targets | mAP | mAP@0.5 | mAP@0.75 |
|:---:|:---:|---|---|---|
| | | 41.7 | 62.8 | 45.6 |
| ✓ | | 42.5 | 63.0 | 46.6 |
| ✓ | ✓ | 44.0 | 64.9 | 47.9 |

Table 4: Ablation of different components. Compared with the model trained by *SiTraining* with normal augmentation, using mixed data augmentation improves performance by 0.8 points. Adding mixed training targets leads to a further improvement of 1.5 points.

## 4.3    Ablation Study

In this section, we validate our design choices on Faster R-CNN with a ResNet-50 backbone.

**Effects of Different Components**    We first study the effects of different components of *MixTraining*. The results are shown in Table 4. Compared to the model trained by *SiTraining* with normally augmented images (41.7 mAP), adopting mixed data augmentation improves mAP by 0.8 points. Further integrating the mixed training targets improves the model by 1.5 points.

**Strategies for Mixing Targets**    We present three different strategies for combining the pseudo boxes and human-annotated ground-truth boxes, and the results of each are shown in Table 5. Compared with the baseline model that uses only human-annotated ground-truth boxes during training, the strategy designed for dealing with the missing label issue brings notable improvements. In comparison, using the strategy that only deals with box localization noise only show a certain advantage. Furthermore, applying both strategies together (i.e. hybrid approach) leads to results that surpass either of the strategies alone.

| strategy | mAP | mAP@0.5 | mAP@0.75 |
|---|---|---|---|
| baseline | 42.5 | 63.0 | 46.6 |
| box loc noise | 42.9 | 64.3 | 47.0 |
| missing label | 43.7 | 64.7 | 48.0 |
| hybrid | 44.0 | 64.9 | 47.9 |

Table 5: Different strategies for combining pseudo boxes and human-annotated ground truth. "baseline" indicates the model trained with mixed data augmentation but without pseudo boxes. "box loc noisy" includes pseudo boxes that address localization noise. "missing label" includes pseudo boxes that deal with missing labels. "hybrid" includes both types of pseudo boxes.

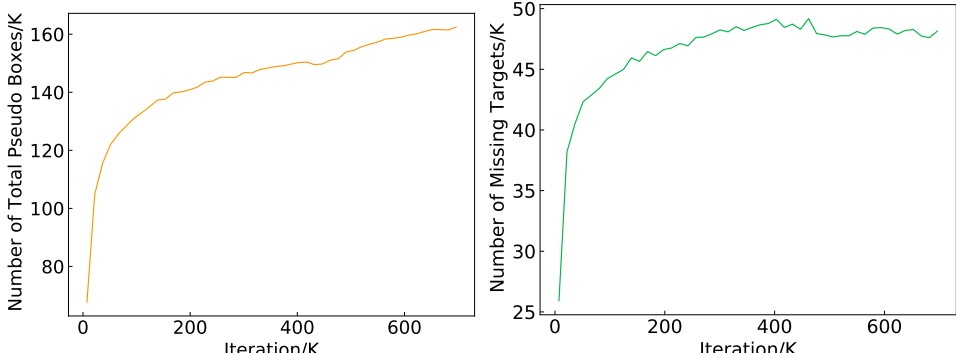

Figure 6: Pseudo box plots. (left) the number of generated pseudo boxes at different training iterations; (right) the number of pseudo boxes retained as missing labels at different training iterations.

In addition, we examine how the number of predicted pseudo boxes and the number of pseudo boxes retained as missing labels changes in the training process. Figure 6 shows the results. As training progresses, both of these quantities increase, indicating that the quality of pseudo boxes improves during training. In Figure 8, some qualitative results of the generated pseudo boxes are displayed.

| Data Aug | Weighted Loss | mAP | mAP@0.5 | mAP@0.75 |
|---|---|---|---|---|
|  |  | 41.7 | 62.8 | 45.6 |
| ✓ |  | 42.1 | 63.2 | 46.3 |
| ✓ | ✓ | 42.5 | 63.0 | 46.6 |

Table 6: Effects of the data augmentation itself and the weighted loss for strongly augmented images.

**Mixed Data Augmentation**   The mixed data augmentation consists of two parts: the mixed data augmentation strategy, and the weighted training loss that includes strong augmentations only on easy training targets. We study the impact of these two parts, and the results are shown in Table 6. Compared with the baseline model learned through *SiTraining* with normal data augmentation, simply applying the mixed data augmentation strategy improves the performance by 0.4 points, and further adopting the weighted loss to alleviate the training difficulty issue for strongly augmented images further improves the performance by 0.4 points.

Figure 7 (left) illustrates how the proportion of easy training targets changes during training. In the early stage, only a small number of training targets are identified as easy targets, so strongly augmented images have little impact on training. As the training progresses, as increasing number of training targets are well fitted and identified as easy targets. At the end of training, more than $50\%$ of the training targets are identified as easy samples and used for strong augmentation. Figure 7 (middle) and (right) illustrate the proportion of objects of different sizes in easy targets and non-easy targets when the model completed the training. In easy target, the large objects is much more than small objects, while the conclusion is opposite in non-easy target.

Since our mixed data augmentation contains both normal augmentation and strong augmentation, we also compared to applying just one type of augmentation, and the results are shown in Table 7. We

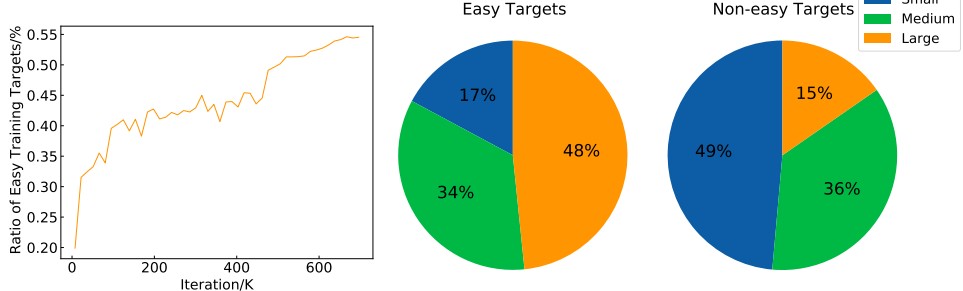

Figure 7: (left) Plot of the proportion of easy targets at different training iterations. (middle) The chart shows the proportion of object of different sizes in easy target. (right) The chart shows the proportion of objects of different sizes in non-easy target.

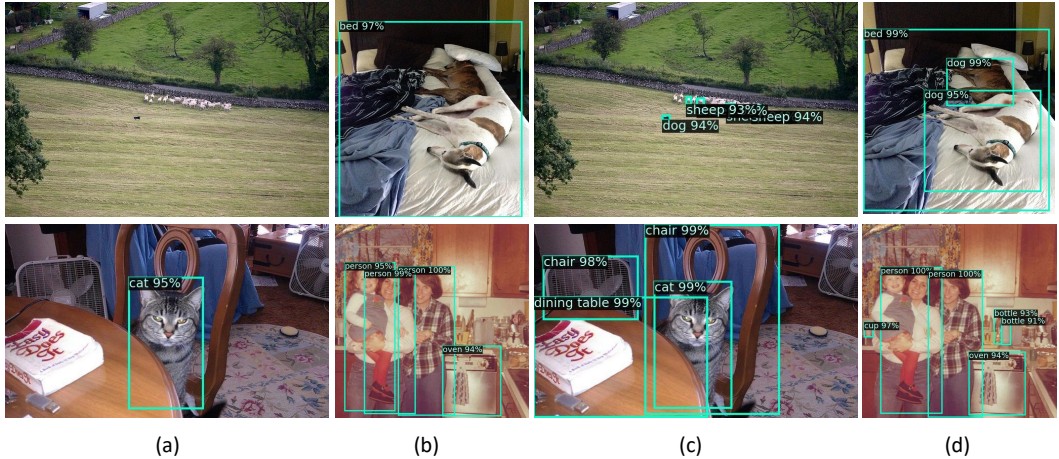

Figure 8: Qualitative results of pseudo boxes. (a) and (b) show the detected pseudo boxes at an early training stage (3k iterations); (c) and (d) show the pseudo boxes generated at the end of training.

| Data Aug | mAP | mAP@0.5 | mAP@0.75 |
|---|---|---|---|
| Normal | 41.7 | 62.8 | 45.6 |
| Strong | 40.7 | 62.2 | 44.4 |
| Mixed | 42.5 | 63.0 | 46.6 |

Table 7: Mixed data augmentation strategy vs. single augmentation strategy.

found that simply adopting strong augmentation in *SiTraining* does not improve performance. On the contrary, it hurts the training process, resulting in worse performance than using normal augmentation. Our mixed data augmentation strategy can alleviate the training difficulty of strong augmentation and improve the performance by 0.8 points.

## 5  Conclusion

In this work, we introduce a new training paradigm, MixTraining, for object detection that can improve the performance of existing detectors for free. Our method consists an enhanced data augmentation that combines different strengths of augmentation and excludes the strong augmentations of certain training targets to reduce training difficulty. In addition, a pseudo box mechanism is introduced to address label noise in human annotation. The experiments conducted on the COCO2017 benchmark verify the effectiveness of MixTraining across various detectors and backbones.

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
