# OpenReview forum: "Bootstrap Your Object Detector via Mixed Training"
_NeurIPS.cc/2021/Conference — NeurIPS 2021 Spotlight_

### Official Review · Reviewer_7Km7 · 2021-07-15

**Rating:** 8
**Confidence:** 5

**Summary:**

This paper introduces a new training paradigm named MixTraining for object detection that can improve the performance of existing detectors for free. The main idea is that MixTraining enhances data augmentation by utilizing augmentations of different strengths. MixTraining is found to bring consistent improvements across various detectors on the COCO dataset. It improves faster R-CNN from 41.7 to 44.0 and Swin-small from 50.9 to 52.8.

**Limitations And Societal Impact:**

The authors adequately addressed the limitations and potential negative societal impact of their work.

**Main Review:**

Strengths:
1) The motivation to perform instance-wise augmentation for different instance is clear and make sense;
2) The method is straightforward and easy to follow;
3) The effectiveness of the method is validated on a strong baseline, which is a 50 + mAp detector on COCO dataset.
4) No further computation cost is found in the proposed method, MixTraining can bring improvement for free.
5) Good experimental results and ablation studies.

Weaknesses:
1) The network has to recognize each instance with different augmentations, which may slow down the training time.

**Time Spent Reviewing:**

5h

---

> ### Author Response · Authors · 2021-08-10
> **Response to Reviewer 7Km7**
>
> Thanks for your constructive comments and your recognition of the solid contribution of this approach.
>
> **Q1: The network has to recognize each instance with different augmentations, which may slow down the training time.**
>
> The additional training cost mainly comes from the process of generating pseudo-labels. A possible solution to reduce the additional training cost is to reduce the frequency of re-generating the pseudo-labels. We take this exploration as our future work.

---

> > ### Comment · Reviewer_7Km7 · 2021-08-25
> > **Thanks for the feedback**
> >
> > Thanks for the author's feedback. I have no more questions and decide to keep my score unchanged.

---

### Official Review · Reviewer_eNWD · 2021-07-16

**Rating:** 7
**Confidence:** 4

**Summary:**

Object detection is a fundamental task in computer vision. In this paper a new paradigm of training, MixTraining, is proposed which utilizes augmentation of different strengths. Either standard augmentation and standard processing is used for an image or strong augmentation is applied (mixed data augmentation) and changed ground truth is used (loss is computed only on simple boxes and mixed training targets is used). As soon as human-annotated bounding boxes often are noisy (missing or inaccurate localizations) the other key ingredient when strong augmentation is applied to an image is to use only confident bounding boxes predicted by the model itself (fixes errors in human annotations) in addition to the confident bounding boxes from the ground truth (making simpler task as soon as strong augmentation is used). This key ingredient is obtained via pseudo-labeling by EMA detector (pseudo boxes). The idea is that this pseudo boxes and bootstrapping from the model itself help to detect images where strong augmentation can be expected to be helpful. MixTraining is shown to consistently improve various detectors on the COCO dataset.

**Limitations And Societal Impact:**

Didn't discussed in the paper. It is referenced to Appendix in the Checklist, however, Appendix was not attached to the submission. Either include in the Appendix and post it or add in the main text.

**Main Review:**

**Originality**

Authors propose a novel paradigm of training for object detection when both normal and strong augmentations are involved and pseudo boxes are used to define how particular box is hard or not to use as a target in strongly augmented sample.  Augmentation, EMA and pseudo labeling are very well known, however, the latest  is more commonly used in semi/self-supervised setting and not for the supervised one to guide how hard or simple to detect particular bounding box. Known techniques are used from a new point of view.

**Quality**

There is no any theoretical analysis and work is purely experimental. However, all experiments are very well defined and looks reasonable and technically strong:
- different backbones and detectors are considered
- careful training schedules are used as long as different models and paradigms have different convergence speed (of course due to stronger augmentation used in MixTraining for example).
- extensive ablations are performed to understand the mixed targets and mixed augmentations dynamics
- extra analysis on pseudo boxes is conducted

However some extra information on choice of crucial MixTraining hyper-parameters is not provided/discussed:
- When is strong augmentation branch involved into training: right at the beginning or after EMA model reaches some good performance? Did authors observe any problem on bootstrapping with pseudo boxes?
- Are pseudo boxes regenerated every update? Does model generate pseudo boxes and strong targets once per N updates for all data?
- What is the dynamics of the parameters from above two bullets?
- Is probability to apply normal and strong  augmentations 0.5? Or do for the update step we apply both normal and strong augmentation for an image and compute Loss_normal + Loss_strong? Do we apply only strong/normal augmentation to the whole batch or mix of normal and strong ones?

Other comments:
- Is prediction score threshold tuned somehow, why is its value chosen to be 0.9?
- Would be great to have experiment also with on-stage detector to have broader experimental proof that method works.
- How were the augmentations and their parameters found for normal and strong augmentation strategies?
- In fig. 5 seems the learning rate decay happens at some point when there is a gap in model performances. Is it true? If yes then the best learning strategy is perform decay when the validation mAP is at maximum. Did authors try this schedule to avoid overfitting in SiTraining earlier to decay learning rate?

**Clarity**

The paper is very well written and organized (no any typos found during reading). Explanations with proper references on original works are well described too. Comparison with existed methods is fully covered in Section 2. All details on the reproduction steps are provided and the new training paradigm is clear described in the text and by Fig. 1.

**Significance**

MixTraining is a novel general framework, which demonstrates ability on bootstrapping and self-improving even with noisy labels like missed or inaccurate localizations. MixTraining does not introduce extra parameters/computations at inference time, while improve leading detectors significantly via the training strategy itself (Faster R-CNN 41.7 -> 44 mAP; Cascade R-CNN 50.9 -> 52.8 mAP). This also opens potential research for semi-supervised learning as soon as bootstrapping via pseudo-labeling by EMA detector is demonstrated to work.

**Update on decision:**
Changing final decision from 6 to 7 as authors addressed all concerns in the below comments.

**Time Spent Reviewing:**

12

---

> ### Author Response · Authors · 2021-08-10
> **Response to Reviewer eNWD**
>
> We thank the reviewer for the constructive comments and would like to address the concerns below.
>
> **Q1: When is strong augmentation branch involved into training: right at the beginning or after EMA model reaches some good performance? Did authors observe any problem on bootstrapping with pseudo boxes?**
>
> The strong augmentation branch is involved at the beginning of training. We also tried to introduce the strong augmentation after the model reaches a good level of performance, but it does not perform better and thus is not adopted in our default settings.
>
> There will be false positives in pseudo boxes. Nevertheless, we empirically find a high threshold of 0.9 will result in very few false positives and works pretty well, and please refer to Q5 for more details.
>
>
> **Q2: Are pseudo boxes regenerated every update? Does model generate pseudo boxes and strong targets once per N updates for all data?**
>
> Yes, we regenerate the pseudo boxes at each training iteration. We did not use the "once per N updates" strategy. It is a good suggestion. Thanks!
>
>
> **Q3: What are the dynamics of the parameters from above two bullets?**
>
> Figure 5 and 7 can partly reflect the training dynamics by the proposed MixTraining approach. It converges slower than that of the SiTraining approach, but can reach a higher accuracy when converged.
>
>
> **Q4: Is probability to apply normal and strong augmentations 0.5? Or do for the update step we apply both normal and strong augmentation for an image and compute Loss\_normal + Loss\_strong? Do we apply only strong/normal augmentation to the whole batch or mix of normal and strong ones?**
>
> In each training iteration, we sample 32 images to form a training batch, of which 16 images use the normal augmentation and the other 16 images use the strong augmentation. The final loss is averaged from these two sub-batches.
>
>
> **Q5: Is prediction score threshold tuned somehow, why is its value chosen to be 0.9?**
> Yes, we tuned this hyper-parameter in a range of 0.8 to 0.95, and found 0.9 works well, the results are shown in following table:
>
> | Threshold | mAP |
> | ------ | ------ |
> |  0.8  |   43.9   |
> |  0.9  |   44.0   |
> |  0.95 |   43.0   |
>
>
> **Q6: Would be great to have experiment also with one-stage detector to have broader experimental proof that method works.**
>
> To address the reviewer's question, we have validated our method using a single-stage detector, RetinaNet. Due to the limited rebuttal period, we are only able to get results with a short training schedule (360K training steps with a batch size of 32). Results are shown in following table:
>
> | Method                            |     mAP        |
> | --------------------------------  | -------------- |
> | SiTraining + RetinaNet (best performance in 180K~540K)           |   41.2         |
> | MixTraining + RetinaNet (360K)    |   41.7 (+0.5)   |
> | SiTraining + Faster R-CNN (best performance in 180K~720k)        |   41.7         |
> | MixTraining + Faster R-CNN (360K) |   42.4 (+0.7)   |
> | MixTraining + Faster R-CNN (720K) |   44.0 (+2.3)   |
>
> Our method improves over a RetinaNet baseline by +0.5 mAP. The gain is similar to that of Faster R-CNN using the same training schedule length (+0.7 mAP).
>
>
> **Q7: How were the augmentations and their parameters found for normal and strong augmentation strategies?**
>
> Our augmentation design mainly refers to RandAug [1] and CutOut [2]. We have empirically removed some transformations that may hurt training to form our normal enhancement, and we use all transformations to form the strong augmentation.
>
> **Q8: In fig. 5 seems the learning rate decay happens at some point when there is a gap in model performances. Is it true? If yes then the best learning strategy is perform decay when the validation mAP is at maximum. Did authors try this schedule to avoid over-fitting in SiTraining earlier to decay learning rate?**
>
> The performance of SiTraining at different total training iterations as shown below:
>
> | Method           | 180K | 360K | 540K | 720K |
> | ---------------- | ---- | ---- | ---- | ---- |
> |SiTraining (Normal)| 41.7 | 41.7 | 40.2 | 36.1 |
> |SiTraining (Strong)| 40.1 | 40.7 | 40.5 | 38.9 |
> |MixTraining       |  -   | 42.4 | -    | 44.0 |
>
> The best results achieved by SiTraining (Normal) and SiTraining (Strong) under different training schedules are worse than our MixTraining.
>
> In particular, for the 180K training schedule, the learning rate decayed at 120K iterations, which is very close to the iteration where SiTraining (Normal) reaches the highest mAP in Figure.5 (left), but its final performance has no significant difference compared to the 360K training setting.
>
>
> **Q9: Didn't discussed in the paper. It is referenced to Appendix in the Checklist, however, Appendix was not attached to the submission. Either include in the Appendix and post it or add in the main text.**
>
> We apologize for the lack of discussion about the limitations and societal impact of this work. We will add the following content in the revision:
>
> **Limitations:** This work studies the mixed-training strategy for only object detection. However, this method has the potential to be applied to other tasks, such as classification and segmentation, which we would like to explore in future work. In addition, this method can benefit from longer training, but at the same time it also increases training costs. How to reduce training time while achieving good performance is another direction worth exploring.
>
> **Societal Impact:** This work focuses on a basic computer vision research problem and has no direct relationship with real societal problems. However, the technology developed in this work may inspire other related research.
>
> ## Reference
>
> [1] Randaugment: Practical automated data augmentation with a reduced search space. Cubuk, et al.
>
> [2] Improved regularization of convolutional neural networks with cutout. DeVries, et al.
>
> [3] Swin transformer: Hierarchical vision transformer using shifted windows. Liu, et al.

---

> > ### Comment · Reviewer_eNWD · 2021-08-23
> > **Changing the paper rating**
> >
> > Dear authors,
> >
> > Thanks a lot for all clarification, details and additional experiments provided in the discussion. As you addressed to all my comments and concerns I change my final decision on the paper rating from 6 to 7.
> >
> > Also regarding different strategies on how often re-generate pseudo-labels: recently there were extensive study of pseudo-labeling in speech recognition when small amount or large amount of labeled data is available and different iterative training strategies were proposed. Possibly you will find them interesting to check and try some ideas or clues in object detection too.
> >
> > - Semi-Supervised ASR by End-to-End Self-Training. Proc. Interspeech 2020, pp.2787-2791.
> > - Iterative Pseudo-Labeling for Speech Recognition. Proc. Interspeech 2020, pp.1006-1010.
> > - slimIPL: Language-model-free iterative pseudo-labeling. 2021. arXiv preprint arXiv:2010.11524, Interspeech 2021
> > - Momentum Pseudo-Labeling for Semi-Supervised Speech Recognition. arXiv preprint arXiv:2106.08922, Interspeech 2021
> > - Kaizen: Continuously improving teacher using Exponential Moving Average for semi-supervised speech recognition. 2021. arXiv preprint arXiv:2106.07759.

---

### Official Review · Reviewer_WPyC · 2021-07-17

**Rating:** 7
**Confidence:** 4

**Summary:**

The study offers MixTraining, an object detection training technique that may be used to improve the performance of existing detectors. The suggested method improves data augmentation by leveraging augmentations of various strengths while eliminating strong augmentations of particular training samples, which could be harmful to training. It addresses localization noise and missing labels in human annotations as well. The efficacy of MixTraining is validated on the COCO dataset using different deep learning models and backbones.

**Limitations And Societal Impact:**

I do not think there is any potential negative societal impact of this work.

**Main Review:**

This study utilizes augmentations of varying strengths while avoiding strong augmentations of certain training samples commendably. Psedo-labeling has been around for a long time. Overall, the authors enhanced a few existing techniques in an effective manner.

The authors demonstrate how to effectively use an augmentation technique and the value of human-annotated bounding boxes to bootstrap an object detector by extending its capabilities. They show that a model/backbone trained using the suggested mechanism outperforms a model/backbone trained using a typical setting.

The manuscript is well-organized, well-written, and easy to follow. The motivation of the study is clear and well-defined. The authors evidently explain how to apply an augmentation strategy to improve an object detector's performance.

The proposed mechanism is useful. The study does a great job of improving the performance of a detector by employing a previously less explored mechanism. However, in my opinion, the contribution of the paper is somewhat incremental which effectively uses the MixTraining technique to improve the capabilities of existing detectors.

To sum up, the authors demonstrate how an augmentation method and the efficacy of human-annotated bounding boxes can be extended beyond a typical training paradigm for object detection. I think this is an interesting approach that could prove beneficial to the research community. The proposed method can also be utilized effectively in real-world applications to enhance the performance of an object detector. Overall, I think this is a good study and am inclined to accept the paper.

**Time Spent Reviewing:**

8

---

> ### Author Response · Authors · 2021-08-10
> **Response to Reviewer WPyC**
>
> Thanks for your valuable feedback. In particular, we appreciate your recognition of the solid contribution of this approach. We also appreciate your comments on the contribution of this work, which inspired us to further explore the application of MixTraining in other tasks.

---

> > ### Comment · Reviewer_WPyC · 2021-08-26
> > **Thanks for the response.**
> >
> > I have checked other reviews and authors' responses. I would like to keep my score unchanged.

---

### Official Review · Reviewer_iVVT · 2021-07-17

**Rating:** 6
**Confidence:** 4

**Summary:**

This paper proposes a method to improve the performance of object detectors with supervised training. The main ideas consist of two parts: (1) The human annotation of bounding boxes is often noisy or incomplete. Adopting a teacher model to generate pseudo boxes and combining them with original annotation can improve the performance of object detectors. (2) Strong augmentation may generate data which degrades the training performance. Adopting a teacher model to select easy targets in the augmented images for training can improve the performance.
In this paper, the author adopts an exponential moving average model as the teacher model. The proposed MixTraining strategy improves the performance (1.6~2.3 mAP) of several object detectors in COCO benchmark.


**Limitations And Societal Impact:**

See above

**Main Review:**

Strengths:
This paper is well written and easy to follow. The idea of adopting teacher model to handle the out of distribution data generated by strong augmentation is simple and effective.
The ablation shows the effectiveness of adopting pseudo boxes in the supervised object detection training.

Weakness:
The proposed method could only address the missing label problem or inaccurate bounding box localization. As for false positive annotation, the proposed method can not handle.
Missing references for data augmentation works which adopt different augmentation policies for different data samples during training.
As claimed in this paper, the author considers strong data augmentation may degrade the performance of object detectors and provide experiments in Table 6. However, as shown in [1] and [2], the searched data augmentation can improve the performance significantly. Adopting the proposed method on [1] or [2] will make the experiments stronger.

[1] Learning Data Augmentation Strategies for Object Detection.
[2] Scale-aware Automatic Augmentation for Object Detection.

Comments:
Could the authors adopt the proposed method for one-stage object detectors? Why do the authors only provide results of two-stage detectors?


**Time Spent Reviewing:**

1

---

> ### Author Response · Authors · 2021-08-10
> **Response to Reviewer iVVT**
>
>
> We thank the reviewer for the constructive comments and would like to address the concerns below.
>
> **Q1: The proposed method could only address the missing label problem or inaccurate bounding box localization. As for false-positive annotation, the proposed method cannot handle it.**
>
> Thank you for pointing this out. Although false-positive annotations are rare in commonly used detection datasets, i.e. COCO dataset, solving this challenge is still a valuable problem for real-world applications, and there is little exploration in existing work. We acknowledge that our method is not designed for this challenge, and we will explore possible solutions in future work: for example, to solve the problem by considering the consistency between predictions and annotations. We will also discuss this problem as part of the limitations of this work in the revision.
>
> **Q2: As claimed in this paper, the author considers strong data augmentation may degrade the performance of object detectors and provide experiments in Table 6. However, as shown in [1] and [2], the searched data augmentation can improve the performance significantly. Adopting the proposed method on [1] or [2] will make the experiments stronger.**
>
> Appropriate design of data augmentations is complementary to our method.  We tried to combine our method with AutoAug by using the AutoAug as the normal augmentation. Due to the short rebuttal period, we are only able to generate results with a short training schedule (360K training steps with a batch size of 32). Since AutoAug does not report results of Faster R-CNN with ResNet-50, we run the AutoAug baseline based on the implementation provided in MMDetection [3] (`mmdet.datasets.pipelines.AutoAugment`). The results are shown in the following table:
>
> | Method      |  Normal Augmentation |     mAP      |
> | ------------| ---------------------| -------------|
> | SiTraining  |  default Normal      |   41.7       |
> | MixTraining |  default Normal      | 42.4 (+0.7)  |
> | SiTraining  |  AutoAug [1]         | 41.5         |
> | MixTraining |  AutoAug [1]         | 42.7 (+1.2)  |
>
> Our method also improves the performance of the AutoAug baseline by 1.2 mAP, which is even larger than for our default normal augmentation.
>
> It is worth noting that the performance of AutoAug is close to our default normal data augmentation. This is because our normal augmentation is stronger than the commonly used weak data augmentation, such as the default augmentation used in MMDetection. In fact, even compared with the result reported in ScaleAware-Aug [2], the performance of our normal data augmentation is also comparable, as shown in the following table:
>
> |                      | MMDetection baseline |SiTraining (default Normal) | SiTraining (AutoAug [1]) | Scale-Aware Aug [2] |
> | -------------------- | -------------------- | ----------------- | ---------------------- | -------------- |
> | Faster R-CNN (Res50) |         40.3         |       41.7        |           41.5         |      41.8      |
>
> The above experiments indicate that our method is compatible with existing data augmentation methods. We will add the related experiments in the revision.
>
>
>
> **Q3: Missing references for data augmentation works which adopt different augmentation policies for different data samples during training.**
>
> Thanks for this suggestion. We will add a discussion in the revision.
>
> **Q4: Could the authors adopt the proposed method for one-stage object detectors? Why do the authors only provide results of two-stage detectors?**
>
> To address the reviewer's question, we have validated our method using a single-stage detector, RetinaNet. Due to the limited rebuttal period, we are only able to get results with a short training schedule (360K training steps with a batch size of 32). Results are shown in the following table. We will add this ablation study in the revision.
>
> | Method                            |     mAP        |
> | --------------------------------  | -------------- |
> | SiTraining + RetinaNet (best performance in 180K~540K)           |   41.2         |
> | MixTraining + RetinaNet (360K)    |   41.7 (+0.5)   |
> | SiTraining + Faster R-CNN (best performance in 180K~720k)        |   41.7         |
> | MixTraining + Faster R-CNN (360K) |   42.4 (+0.7)   |
> | MixTraining + Faster R-CNN (720K) |   44.0 (+2.3)   |
>
> Our method improves over a RetinaNet baseline by +0.5 mAP. The gain is similar to that of Faster R-CNN using the same training schedule length (+0.7 mAP).
>
> ## References
>
> [1] Learning Data Augmentation Strategies for Object Detection.  Zoph, Barret, et al.
>
> [2] Scale-aware Automatic Augmentation for Object Detection. Chen, Yukang, et al.
>
> [3] MMDetection: Open MMLab Detection Toolbox and Benchmark. Chen, et al.

---

### Decision · Program_Chairs · 2021-09-27

**Decision:**

Accept (Spotlight)

**Comment:**

The paper presents a new training procedure for object detection that relies on a teacher model and data augmentation. The approach adds a loss that is computed on a strongly augmented image but only on boxes that are above a threshold for the teacher (exponential moving average) model on the raw image. It is quite simple and generic, and provides a significant boost in final performance. The experiments are conducted on COCO with strong standard models (Faster R-CNN and Cascade-R-CNN) models. The reviewers have made constructive observations and all agree that this is a good paper. Overall, the paper is a significant contribution and is a welcome addition to the program of NeurIPS.